# Lung Transplant Immunomodulation with Genetically Engineered Mesenchymal Stromal Cells—Therapeutic Window for Interleukin-10

**DOI:** 10.3390/cells13100859

**Published:** 2024-05-17

**Authors:** Antti I. Nykänen, Andrea Mariscal, Allen Duong, Aadil Ali, Akihiro Takahagi, Xiaohui Bai, Guan Zehong, Betty Joe, Mamoru Takahashi, Manyin Chen, Hemant Gokhale, Hongchao Shan, David M. Hwang, Catalina Estrada, Jonathan Yeung, Tom Waddell, Tereza Martinu, Stephen Juvet, Marcelo Cypel, Mingyao Liu, John E. Davies, Shaf Keshavjee

**Affiliations:** 1Latner Thoracic Research Laboratories, Toronto General Hospital Research Institute, University Health Network, Toronto, ON M5G 1L7, Canada; antti.nykanen@helsinki.fi (A.I.N.); andrea.mariscal@uhn.ca (A.M.); allen.duong@mail.utoronto.ca (A.D.); aadil.ali@uhn.ca (A.A.); akihiyot@gmail.com (A.T.); xhbai@hotmail.com (X.B.); zehong.guan@uhnresearch.ca (G.Z.); betty.joe@uhnresearch.ca (B.J.); mt10947@kuhp.kyoto-u.ac.jp (M.T.); manyin.chen@uhn.ca (M.C.); hemant.gokhale@uhn.ca (H.G.); hongchao.shan@uhn.ca (H.S.); jonathan.yeung@uhn.ca (J.Y.); tom.waddell@uhn.ca (T.W.); tereza.martinu@uhn.ca (T.M.); stephen.juvet@uhn.ca (S.J.); marcelo.cypel@uhn.ca (M.C.); mingyao.liu@utoronto.ca (M.L.); 2Institute of Medical Science, Temerty Faculty of Medicine, University of Toronto, Toronto, ON M5S 1A8, Canada; 3Division of Thoracic Surgery, Department of Surgery, Temerty Faculty of Medicine, University of Toronto, Toronto, ON M5T 1P5, Canada; 4Toronto Lung Transplant Program, Ajmera Transplant Centre, University Health Network, Toronto, ON M5G 2N2, Canada; 5Laboratory Medicine and Molecular Diagnostics, Sunnybrook Health Sciences Centre, University of Toronto, Toronto, ON M4N 3M5, Canada; david.hwang@sunnybrook.ca; 6Tissue Regeneration Therapeutics, Toronto, ON M5G 1N8, Canada; catalina@verypowerfulbiology.com (C.E.); jed.davies1063@gmail.com (J.E.D.); 7Division of Respirology, Department of Medicine, Temerty Faculty of Medicine, University of Toronto, Toronto, ON M5S 1A8, Canada; 8Institute of Biomedical Engineering, University of Toronto, Toronto, ON M5S 3G9, Canada

**Keywords:** lung transplantation, cell therapy, gene therapy, ex vivo lung perfusion, interleukin-10

## Abstract

Lung transplantation results are compromised by ischemia–reperfusion injury and alloimmune responses. Ex vivo lung perfusion (EVLP) is used to assess marginal donor lungs before transplantation but is also an excellent platform to apply novel therapeutics. We investigated donor lung immunomodulation using genetically engineered mesenchymal stromal cells with augmented production of human anti-inflammatory hIL-10 (MSCs^IL-10^). Pig lungs were placed on EVLP for 6 h and randomized to control (*n* = 7), intravascular delivery of 20 × 10^6^ (*n* = 5, low dose) or 40 × 10^6^ human MSCs ^IL-10^ (*n* = 6, high dose). Subsequently, single-lung transplantation was performed, and recipient pigs were monitored for 3 days. hIL-10 secretion was measured during EVLP and after transplantation, and immunological effects were assessed by cytokine profile, T and myeloid cell characterization and mixed lymphocyte reaction. MSC^IL-10^ therapy rapidly increased hIL-10 during EVLP and resulted in transient hIL-10 elevation after lung transplantation. MSC^IL-10^ delivery did not affect lung function but was associated with dose-related immunomodulatory effects, with the low dose resulting in a beneficial decrease in apoptosis and lower macrophage activation, but the high MSC^IL-10^ dose resulting in inflammation and cytotoxic CD8^+^ T cell activation. MSC^IL-10^ therapy during EVLP results in a rapid and transient perioperative hIL-10 increase and has a therapeutic window for its immunomodulatory effects.

## 1. Introduction

Lung transplantation can return a patient with severe end-stage lung disease to normal life. However, the median survival of lung transplant recipients is currently limited to about 7 years [1,2,3], mainly due to primary graft dysfunction, acute rejection and chronic lung allograft dysfunction [4,5,6]. These factors involve an interplay between innate and adaptive immunity [4,7,8]. Novel donor lung immunomodulatory therapeutic strategies could limit early pathological immune responses and improve long-term survival.

Ex vivo lung perfusion (EVLP) is an innovative approach to assessing donor lung quality before lung transplantation [9]. EVLP has safely increased the number of lung transplants, as marginal donor lungs with stable lung function during EVLP can be transplanted with excellent early [10], mid-term and long-term results [11]. EVLP is also an ideal platform for novel therapeutics as, for example, gene and cell therapies could be administered to the donor lung during the ex vivo perfusion period [12,13,14].

Gene therapy has been successfully used in experimental transplant and EVLP models [13,15,16]. However, as it takes several hours to achieve therapeutic levels of the transgenic protein, donor lung gene therapy could miss the relatively narrow therapeutic time frame for perioperative graft protection [15,16]. Various cell therapies such as mesenchymal stromal cells (MSCs) have also been applied during EVLP [13,17]. MSCs elicit immunomodulatory effects primarily through paracrine mechanisms [17,18,19], and tailored genetic engineering of MSCs further expands their therapeutic potential [14,18]. We recently generated genetically modified MSCs that have augmented anti-inflammatory interleukin-10 production (MSCs^IL-10^) [20]. Treatment of human lungs rejected from clinical transplantation with MSCs^IL-10^ during EVLP efficiently elevated IL-10 levels within minutes [20]. 

Here, we used a translational pig EVLP and lung transplant model to determine the potential protective and immunomodulatory effects of MSC^IL-10^ therapy, with two different MSC^IL-10^ doses to investigate dose-related effects. Administration of MSCs^IL-10^ during EVLP increased IL-10 levels rapidly and dose-dependently and resulted in transient IL-10 elevation after lung transplantation. Interestingly, the two MSC^IL-10^ doses induced divergent immunological effects, with the low MSC^IL-10^ dose resulting in decreased apoptosis and lower lung macrophage activation, while the high MSC^IL-10^ dose caused increased inflammation and cytotoxic CD8^+^ T cell activation.

## 2. Materials and Methods

Human umbilical cord MSCs with augmented human IL-10 production (hIL-10) were genetically engineered by adenoviral transduction and then cryopreserved. hIL-10 secretion was confirmed in vitro. Pig double lungs subjected to 24 h cold ischemia were connected to clinical-grade EVLP (using the Toronto EVLP technique) [21] for 6 h and randomized to control (*n* = 7), or to receive 20 × 10^6^ (*n* = 5) or 40 × 10^6^ cryopreserved MSCs^IL-10^ (*n* = 6) through the pulmonary artery in a blinded manner. Lung function parameters, perfusate, lung tissue and bronchoalveolar lavage (BAL) samples were collected during EVLP. After EVLP, the left lung was transplanted into a recipient pig that received methylprednisolone and cyclosporine A background immunosuppression [22]. Three days after transplantation, graft evaluation was performed. Blood, tissue and BAL samples were collected for biochemical, histological and immunological analysis.

Detailed *n*-values are given in Appendix A and in the respective Figure Legends. Please see the online supplement for detailed methods. All animals received humane care in compliance with the “Principles of Laboratory Animal Care,” formulated by the National Society for Medical Research, and The Guide for the Care of Laboratory Animals, published by the National Institutes of Health. The experimental protocol was approved by the Animal Care Committee of the Toronto General Hospital Research Institute (AUP 2999).

## 3. Results

### 3.1. Cryopreserved MSCs^IL-10^ Secrete hIL-10 Rapidly after Thawing In Vitro

MSCs^IL-10^ were pre-engineered by adenoviral vectors encoding hIL-10 and cryopreserved (Figure 1A). Two different MSC^IL-10^ doses were used in subsequent EVLP and lung transplant studies (low dose 20 × 10^6^ MSC^IL-10^; high dose 40 × 10^6^ MSC^IL-10^), and they efficiently produced hIL-10 within minutes in vitro after thawing (Figure 1B) and had excellent (>90%) viability (Appendix A).

To investigate the potential immunomodulatory effects of MSC^IL-10^ therapy during lung transplantation, a pig EVLP and transplant model was used (Figure 1C). MSCs^IL-10^ were administered during EVLP through the pulmonary artery (Figure 1D) 71 ± 4 min after MSC^IL-10^ thawing. Measurements of the supernatant of the MSC^IL-10^ preparation indicated the generation of 0.57 ± 0.14 and 1.66 ± 0.41 μg of preformed hIL-10 (mean ± SD) by the 20 × 10^6^ and 40 × 10^6^ MSCs^IL-10^, respectively, that was thus co-injected into the EVLP circuit together with the MSCs^IL-10^ (Figure 1E).

### 3.2. MSC^IL-10^ Delivery during EVLP Results in Rapid hIL-10 Elevation in Perfusate, Lung Tissue and Airways 

To investigate the kinetics of hIL-10 production during EVLP, pig double lungs with 24 h cold ischemia were placed on the EVLP system, and 20 × 10^6^ or 40 × 10^6^ MSCs^IL-10^ were administered to the perfusate entering the pulmonary artery (Figure 2A) after the first hour on EVLP (Appendix A). Perfusate hIL-10 levels increased within 5 min after cell administration to 492 ± 144 and 1022 ± 390 pg/mL in the 20 × 10^6^ and 40 × 10^6^ MSC^IL-10^ groups, respectively (Figure 2B), likely resulting from the co-administrated preformed hIL-10 (Figure 1E) being distributed into the EVLP circuit. Subsequently, perfusate hIL-10 concentration continued to increase steadily and dose-dependently and reached very high levels of 6068 ± 1384 and 17,194 ± 5573 pg/mL in the 20 × 10^6^ and 40 × 10^6^ MSC^IL-10^ groups, respectively, 6 h after EVLP start (Figure 2B).

A corresponding increase in lung hIL-10 was observed with tissue concentrations reaching 37 ± 16 and 76 ± 44 pg per mg of lung protein at the end of EVLP in the 20 × 10^6^ and 40 × 10^6^ MSC^IL-10^ groups, respectively (Figure 2C), far exceeding the levels achieved in previous adenoviral gene therapy studies [15]. Similar hIL-10 levels were detected in different anatomical lung areas (Figure 2D), and an analysis of BAL samples indicated an increase in hIL-10 in the airway compartment (Figure 2E).

### 3.3. MSC^IL-10^ Therapy during EVLP Results in Transient hIL-10 Increase after Lung Transplantation

We used the pig single-lung transplant model to determine MSC^IL-10^ function and fate after transplantation. After finishing the 6 h EVLP, the left lung was transplanted into a recipient pig that was followed for 3 days (Figure 2F and Appendix A). MSC^IL-10^ treatment resulted in a dose-dependent early hIL-10 peak with recipient plasma levels reaching 261 ± 68 and 723 ± 276 pg/mL 2 h after transplant reperfusion in 20 × 10^6^ and 40 × 10^6^ MSC^IL-10^ groups, respectively (Figure 2G). Subsequently, recipient plasma hIL-10 levels declined to low levels within 1 to 2 days after transplantation and were undetectable at 3 days (Figure 2G). Similarly, lung tissue hIL-10 levels increased 1 h after transplant reperfusion but were not measurable from the transplant, or from the contralateral native lung, at 3 days (Figure 2H). BAL hIL-10 levels were also undetectable 3 days after transplantation (Figure 2E).

To confirm the presence of hIL-10 in the lungs at the mRNA level, lung samples were assessed with qRT-PCR using human-specific IL-10 primers. Mirroring the hIL-10 protein results (Figure 2C,H), lung hIL-10 mRNA levels were increased at the end of EVLP, and at 1 h after transplant reperfusion, and were undetectable 3 days after transplantation (Figure 2I). Also, human vascular endothelial growth factor (hVEGF) mRNA levels from the same lung samples were measured. Low, but detectable, hVEGF mRNA levels were observed (Figure 2J), suggesting that, in addition to the profoundly augmented hIL-10 production, MSCs^IL-10^ also secrete other paracrine factors, but at far lower levels.

### 3.4. MSCs^IL-10^ Delivered during EVLP Are Transiently Retained in the Lung

We next used human (HLA Class I and ALU)- and transgene (FLAG tag attached to the hIL-10 transgene)-specific markers, and multiplex fluorescent staining, to identify MSCs^IL-10^ in the lungs 6 h after EVLP start. HLA Class I^+^ALU^+^FLAG^+^ cells were identified (Figure 2K), indicating that human cells with hIL-10 transgene expression were retained in the pig lung. MSCs^IL-10^ were relatively large (Figure 2K,L), resided in duplicate or as single cells (Figure 2M,N), and were localized mainly to the lung interstitium (Figure 2K,N) and occasionally to the alveolar space (Figure 2L,M). In addition, ALU PCR of lung samples and various recipient tissues was performed to detect human genomic DNA. ALU DNA was present in the lung at the end of EVLP, and was undetectable in the transplanted lung, contralateral native lung and all other non-transplant tissues of the recipient 3 days after transplantation (Figure 2O).

### 3.5. MSCs^IL-10^ Do Not Affect Lung Function during EVLP, but Low-Dose MSC^IL-10^ Treatment Decreases Apoptosis

To determine the effect of MSC^IL-10^ treatment on lung function, double lungs with 24 h cold ischemia were assigned to control, or pulmonary artery delivery of 20 × 10^6^ or 40 × 10^6^ MSCs^IL-10^, in a blinded and randomized manner. The baseline 1 h parameters, obtained before MSC^IL-10^ administration, were similar in all groups (Appendix A). No significant differences were detected in pulmonary vascular resistance (Figure 3A), lung oxygenation (Figure 3B), peak airway pressure (Figure 3C) or dynamic compliance (Figure 3D) after the administration of 20 × 10^6^ or 40 × 10^6^ MSCs^IL-10^. Additionally, the metabolic conditions in all groups remained stable during EVLP (Appendix A), and no significant differences were found in markers for vascular permeability (Figure 3E–G). Analysis of lung apoptosis 6 h after EVLP start indicated that treatment with 20 × 10^6^ MSCs^IL-10^ significantly decreased TUNEL^+^ cell density in the lung alveolar compartment, whereas large airway epithelium apoptosis was infrequent in all groups (Figure 3H–K and Appendix A).

### 3.6. MSC^IL-10^ Treatment Does Not Affect Lung Function after Transplantation

To evaluate the potential effects of MSCs^IL-10^ on post-transplant lung function, pulmonary parameters were collected after transplantation, and detailed analysis of transplanted graft oxygenation, ventilatory function, pulmonary artery pressures and histology were obtained immediately and 3 days after transplantation. MSC^IL-10^ treatment did not change early lung function, determined by selective graft upper and lower lobe P/F-ratios 1 h after reperfusion (Figure 4A), by the time to recipient extubation, or weaning from supplemental oxygen (Figure 4B). During the postoperative phase, no lung-related adverse events occurred, all recipients remained on room air and systemic P/F ratios remained stable (Figure 4C). Detailed lung evaluation 3 days after transplantation indicated no differences in transplant P/F ratios (Figure 4D), pulmonary artery pressures (Figure 4E) or ventilation parameters (Figure 4F,G). Additionally, histological examination indicated similar levels of acute lung injury in all groups (Figure 4H), and no differences in macroscopic lung appearance were detected between groups (Appendix A).

### 3.7. High-Dose MSC^IL-10^ Treatment Results in a Pro-Inflammatory Response

To explore possible immunological effects of the MSC^IL-10^ treatment, various inflammatory cytokines were measured. A modest gradual elevation in perfusate pro-inflammatory cytokines was seen in the control and 20 × 10^6^ MSC^IL-10^ groups. In contrast, treatment with 40 × 10^6^ MSCs^IL-10^ resulted in a significant increase in perfusate IL-1β (Figure 5A), IL-6 (Figure 5B), IL-8 (Figure 5C) and IL-10 levels (Figure 5D), measured by pig-specific cytokine ELISA.

Next, qRT-PCR was used to determine lung inflammatory profiles at various time points of the experiment. In the control group, a general increase in especially IL-6, IL-8 and IL-17 mRNA was observed at 6 h after EVLP start (Figure 5E), and at 1 h after transplant reperfusion, compared to pre-EVLP levels (Figure 5F). Additionally, IL-1α, IL-1β, IL-2, IL-4, tumor necrosis factor (TNF)α and interferon (IFN)α mRNA levels were prominent 3 days after transplantation (Figure 5G). Compared to the control group, treatment with 20 × 10^6^ MSC^IL-10^ led to increased IFNγ mRNA levels 6 h after EVLP start (Figure 5E) and decreased IL-8 mRNA 1 h after transplantation (Figure 5F). In contrast, treatment with 40 × 10^6^ MSCs^IL-10^ resulted in an altered lung cytokine profile with elevation of IFNα and granzyme B mRNA 1 h after reperfusion (Figure 5F), and a significant upregulation of IL-4, TNFα, IFNα, IFNγ and perforin mRNA 3 days after transplantation (Figure 5G).

### 3.8. High-Dose MSC^IL-10^ Therapy Increases Cytotoxic CD8^+^ T Cell Response

Based on the unexpected stimulatory effects of 40 × 10^6^ MSC^IL-10^ treatment on lung IFNγ, perforin and granzymes (Figure 5F,G), and previous studies showing that high levels of IL-10 activate cytotoxic CD8^+^ T cells [23,24,25,26], we next characterized the lung allograft T cell composition 3 days after transplantation (Figure 6 and Appendix A). Flow cytometry of pulmonary immune cells showed that, in contrast to 20 × 10^6^ MSC^IL-10^, treatment with 40 × 10^6^ MSCs^IL-10^ increased lung CD3^+^ (Figure 6A), CD4^+^ (Figure 6B) and CD8^+^ T cell numbers (Figure 6C) compared to the control group. Furthermore, T cell subtype analysis revealed that 40 × 10^6^ MSCs^IL-10^ specifically elevated the number of lung CD45RO^-^CD62L^+^ central memory CD4^+^ T cells (Figure 6D), CD45RO^+^CD62L^+^ naïve CD8^+^ T cells and IFNγ^+^ cytotoxic CD8^+^ T cells (Figure 6E).

### 3.9. Low-Dose MSC^IL-10^ Therapy Decreases Activated Macrophages in the Lung

As the inhibition of macrophage activation has been described as one of the main mechanisms for the anti-inflammatory effects of IL-10 [27,28], we also analyzed the lung myeloid cell composition (Figure 7 and Appendix A). No effects on lung neutrophils (Figure 7A), CD16^+^ monocytes (Figure 7B), CD163^+^ macrophages (Figure 7C) or natural killer (NK) cells (Figure 7D) were found. However, macrophage subtype analysis revealed that treatment with 20 × 10^6^ MSCs^IL-10^ decreased the number of SLAII^-^DR^+^CD163^mid^SSC^low^-activated macrophages (Figure 7F).

### 3.10. High-Dose MSC^IL-10^ Therapy Increases Immunity Early after Lung Transplantation

Finally, we performed mixed lymphocyte reaction assays to determine the overall effect on early alloimmunity. MSC^IL-10^ treatment did not affect CD4^+^ cell proliferation (Figure 8A and Appendix A). However, in contrast to 20 × 10^6^ MSC^IL-10^, treatment with 40 × 10^6^ MSC^IL-10^ increased CD8^+^ cell proliferation compared to the control group (Figure 8B,C). Interestingly, this CD8^+^ cell proliferation occurred irrespective of the amount of stimulator cells present, suggesting that the immune activation related to the high MSC^IL-10^ dose was mainly alloantigen-independent.

## 4. Discussion

We combined cell and gene therapies and applied genetically engineered MSCs^IL-10^ with augmented hIL-10 production in translational large-animal EVLP and lung transplant models. Administration of MSCs^IL-10^ in a clinically feasible way resulted in a rapid, efficient, and dose-dependent hIL-10 increase during EVLP and was followed by a transient hIL-10 elevation after lung transplantation. MSC^IL-10^ treatment did not affect perioperative lung function, but a comparison of two different MSC^IL-10^ doses showed divergent, dose-related immunological effects, with the low MSC^IL-10^ dose leading to decreased apoptosis and lower macrophage activation, and the high MSC^IL-10^ dose resulting in profound pro-inflammatory effects and CD8^+^ cytotoxic T cell activation.

EVLP is an excellent platform for advanced therapeutics aiming for graft protection, repair and immunomodulation [12]. Here, we administered genetically modified MSCs^IL-10^ during EVLP, and, in order to facilitate potential future translation to clinical trials, used pre-modified and cryopreserved MSCs^IL-10^, clinical-grade EVLP and a large-animal transplant model. Similar to previous studies, MSC administration through the pulmonary artery during EVLP resulted in MSC retainment in the lung [14,17,20]. Additionally, our present results provide important information about the subsequent fate of the MSCs^IL-10^ after lung transplantation, as they were still present 1 h after transplant reperfusion, but were undetectable in the transplanted graft and all recipient tissues 3 days after operation. Similarly, transgenic hIL-10 was elevated early after reperfusion, but decreased to low levels 1 to 2 days after transplantation. These results indicate that the viability of MSCs^IL-10^ after lung transplantation is limited. This is also supported by previous findings showing that live MSCs are detected in the lung up to 1 day after intravenous injection, but are then rapidly phagocytosed [29,30]. Considering the kinetics of the transgenic hIL-10 in our study, therapy with genetically modified MSCs during EVLP would ideally target the early perioperative phase of lung transplantation, and could be thus potentially used to counteract ischemia–reperfusion injury and early alloimmune activation. 

Importantly, we found that administration of 20 × 10^6^ MSCs^IL-10^ during EVLP resulted in beneficial effects as it decreased lung apoptosis, lung IL-8 mRNA and macrophage activation. These results are in line with the generally known anti-inflammatory properties of IL-10 [28,31], and with the protective roles of IL-10 in lung ischemia–reperfusion [32,33] and transplantation [15,16]. As alveolar macrophages are critical regulators of lung ischemia–reperfusion injury [33,34,35], and as MSC^IL-10^ treatment increased hIL-10 in the alveolar space and decreased the number of activated macrophages, it is possible that alveolar macrophages mediated the beneficial effects of the 20 × 10^6^ MSC^IL-10^ treatment. However, it is also possible that other IL-10 effects, and MSC effects independent of IL-10, contributed to the observed findings. First, in addition to controlling inflammation, IL-10 is also known to possess non-classical effects on epithelial proliferation and tissue repair [28]. Second, besides the augmented hIL-10 production, conventional paracrine mediators secreted by the MSC^IL-10^ may have participated in the immunomodulation [17,18,19]—a possibility supported by our finding of human VEGF mRNA in the treated pig lungs. Third, as the viability of the MSC^IL-10^ after lung transplantation was limited, MSC apoptosis and phagocytosis may have affected the phenotype of antigen-presenting cells and resulted in changes in adaptive immunity [30,36].

Unexpectedly, the immunological effects of the high MSC^IL-10^ dose were significantly different, as treatment with 40 × 10^6^ MSCs^IL-10^ resulted in the elevation of pro-inflammatory cytokines and the activation of CD8^+^ cytotoxic T cells. While this finding is contrary to the traditional anti-inflammatory characteristics of IL-10 [28,31], it is in line with the reported effects of high-dose IL-10 on CD8^+^ cytotoxic T cells [25,37]. Specifically, several studies have demonstrated that high levels of IL-10 context-dependently induce IFN-γ and activate cytotoxic CD8^+^ T cells, resulting in the upregulation of perforin and granzymes [26,38,39,40,41,42]. These findings have important implications for anti-tumor immunity and, correspondingly, high doses of pegylated IL-10 have been used in preclinical experiments, and in clinical trials in cancer patients [25,26,40]. As we found a profound upregulation of TNF-α, IFN-γ, perforin and granzyme B—prototype cytokines and mediators of cytotoxic cells [25,28]—in the transplanted lungs, and an increase in the number of IFNγ^+^CD8^+^ T cells, it is highly probable that the high-dose MSC^IL-10^ treatment activated cytotoxic T cells. Although the lung NK cell composition remained unchanged, cytotoxic effects on NK cells [43,44] and stimulatory effects on B cells could also have occurred [45].

Overall, our findings highlight the complex and divergent immunomodulatory effects of IL-10 and indicate that there is a therapeutic window for IL-10 in solid organ transplantation: low-to-moderate IL-10 levels have anti-inflammatory effects, probably achieved through macrophages, but very high levels may result in increased cytotoxicity after transplantation [46]. When considering IL-10 therapy in solid organ transplantation, it is therefore essential to investigate dose-related effects, and to determine optimal IL-10 levels and kinetics after IL-10 administration. In our study, low-dose MSC^IL-10^ treatment increased perfusate IL-10 levels to 6 ng/mL and lung IL-10 levels to 37 pg/mg of lung protein during EVLP, while the respective IL-10 levels reached 17 ng/mL and 76 pg/mg of lung protein after high-dose MSC^IL-10^ treatment. Interestingly, in a previous study using human lungs, beneficial anti-inflammatory effects were achieved with adenoviral IL-10 therapy, resulting in lung IL-10 levels of approximately 30 pg/mg of lung protein [15], which is very close to the lung IL-10 levels in our low-dose MSC^IL-10^ group. On the other hand, treatment of cancer patients with daily subcutaneous PEGylated IL-10 resulted in immune and CD8^+^ T cell activation and serum IL-10 levels of 18.9 ng/mL [26], which is very similar to the perfusate IL-10 levels in our high-dose MSC^IL-10^ group. Although the effects of IL-10 are context-dependent, the circulating and tissue levels of IL-10 in our treatment groups may offer some guidance for future studies that aim for anti-inflammatory and protective effects with lower IL-10 dose, or for cytotoxic effects with higher IL-10 levels.

## 5. Conclusions

We leveraged EVLP as a platform to apply advanced cell and gene therapies for donor lung immunomodulation. MSC^IL-10^ treatment resulted in rapid hIL-10 upregulation during EVLP that was followed by a transient hIL-10 elevation after transplantation. MSC^IL-10^ treatment resulted in divergent dose-dependent immunomodulatory effects, indicating that the immunological effects of IL-10 in solid organ transplantation are complex, and that MSC^IL-10^ therapy during EVLP has a therapeutic window that should be optimized.

## Figures and Tables

**Figure 1 cells-13-00859-f001:**
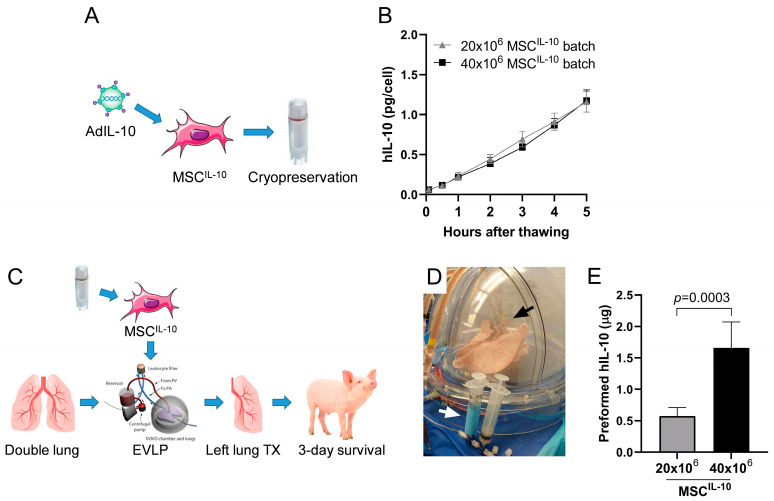
Study design and rapid IL-10 production of cryopreserved MSCs^IL-10^ in vitro. (**A**) Human umbilical cord MSCs were genetically engineered for augmented human IL-10 production (MSCs^IL-10^) and cryopreserved. (**B**) In vitro bench testing of the cryopreserved MSC^IL-10^ batches for hIL-10 production (54 × 10^3^ cells from the 20 × 10^6^ MSC^IL-10^ group, *n* = 5; or from the 40 × 10^6^ MSC^IL-10^ group, *n* = 3). (**C**) Pig double lungs subjected to 24 h cold ischemia were connected to EVLP for 6 h and randomized to control (*n* = 7), or pulmonary artery delivery of 20 × 10^6^ (*n* = 5) or 40 × 10^6^ cryopreserved MSCs^IL-10^ (*n* = 6). Subsequently, the left lung was transplanted into a recipient pig that received methylprednisolone and cyclosporine A background immunosuppression, and was sacrificed 3 days after transplantation. (**D**) A clinical-grade Toronto ex vivo lung perfusion system was used for the EVLP experiments, and MSCs^IL-10^ were administered to the EVLP circuit in a blinded and randomized manner with a syringe (white arrow) and a tubing extension was attached to the pulmonary artery cannula (black arrow). (**E**) Analysis of preformed hIL-10 in MSC^IL-10^ supernatant taken immediately prior to MSC^IL-10^ administration. Data mean ± standard deviation and analyzed with (**B**) two-way ANOVA using Šidák correction for multiple comparisons or (**E**) 2-tailed Student’s *t*-test. Ad, adenovirus; EVLP, ex vivo lung prefusion; hIL-10, human interleukin-10; MSC, mesenchymal stromal cell; TX, transplantation.

**Figure 2 cells-13-00859-f002:**
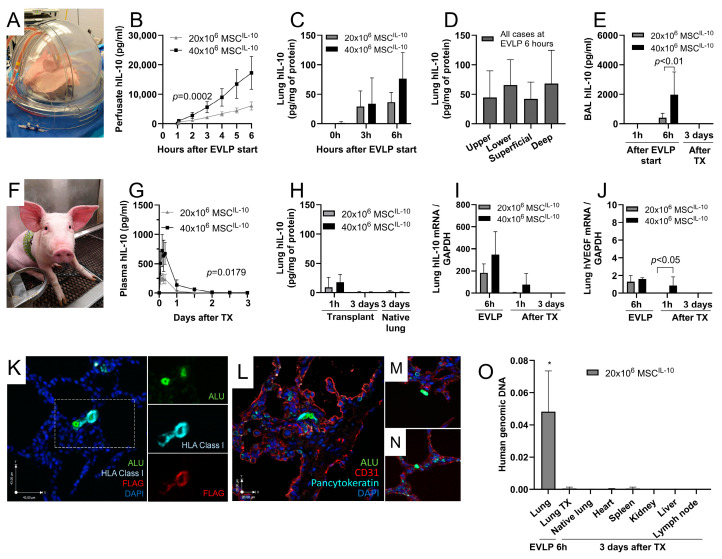
MSC^IL-10^ treatment during EVLP results in rapid dose-dependent hIL-10 elevation that decreases after lung transplantation. Pig double lungs subjected to 24 h cold ischemia were (**A**) connected to EVLP for 6 h, 20 × 10^6^ (*n* = 5) or 40 × 10^6^ (*n* = 6) MSCs^IL-10^ were administered, and hIL-10 was measured using ELISA. (**B**) After MSC^IL-10^ administration, EVLP perfusate hIL-10 levels were rapidly and dose-dependently elevated within minutes and continued to steadily increase for the duration of EVLP. (**C**) Lung tissue hIL-10 protein levels increased during EVLP, and (**D**) no significant differences were detected between samples taken from the upper or lower lobes, or from superficial or deep areas of the lung. (**E**) BAL hIL-10 was dose-dependently elevated 6 h after EVLP start. (**F**) After EVLP, the left lung was transplanted to a recipient pig that received methylprednisolone and cyclosporine A immunosuppression. (**G**) Recipient plasma hIL-10 protein levels peaked dose-dependently during the first 4 h after transplant reperfusion and declined to low levels within 1–2 days after transplantation. (**H**) Lung tissue hIL-10 protein levels were elevated 1 h after lung transplant reperfusion but were undetectable at 3 days. (**I**) Quantitative RT-PCR with human-specific primers of lung samples showed markedly increased hIL-10 mRNA expression at the end of EVLP, lower hIL-10 mRNA levels 1 h after transplant reperfusion and undetectable levels 3 days after transplantation. (**J**) Quantitative RT-PCR for human VEGF mRNA revealed low, but detectable levels of human VEGF mRNA in the lung 6 h after EVLP and 1 h after transplantation. (**K**) Lung tissue multiplex immunofluorescent and fluorescent in situ hybridization staining with human- (HLA Class I and ALU) and transgene-specific markers (FLAG tag; attached to the hIL-10 transgene) detected human MSCs^IL-10^ in the pig lung 6 h after EVLP start. (**K**) MSCs^IL-10^ co-expressed ALU, HLA Class I and FLAG, were relatively large, and (**K**,**L**) resided in duplicate or as (**M**,**N**) single cells. (**L**) Co-staining with endothelial (CD31) and epithelial (pan cytokeratin) markers indicated that MSCs^IL-10^ mainly localized in the (**K**) lung interstitium and occasionally in the (**L**) alveolar space. (**O**) Quantitative PCR with ALU primers indicated the presence of human cells in the pig lungs at the end of EVLP, and 1 h after lung transplantation, but did not detect human genomic DNA in the transplant, or in any non-transplant recipient tissues 3 days after lung transplantation. Data mean ± standard deviation and analyzed by two-way ANOVA, or mixed-effect analysis in cases of missing data, using Šidák correction for multiple comparisons (**B**,**C**,**E**,**G**–**J**), or by one-way ANOVA with pairwise comparisons of the upper and lower lobes, or of superficial and deep location of MSC^IL-10^ 20 × 10^6^ and 40 × 10^6^ samples (**D**), or comparing other tissues to lung transplant (**O**). Results relative to GAPDH housekeeping gene (**I**,**J**). Quantitative PCR ng of human ALU DNA per 500 ng of total DNA, and * *p* < 0.0001 compared to all other tissues (**O**). Insets show magnified single-channel fluorescent images of the dashed area (**K**). BAL, bronchoalveolar lavage; EVLP, ex vivo lung perfusion; hIL-10, human interleukin-10; MSC, mesenchymal stromal cell; TX, transplantation.

**Figure 3 cells-13-00859-f003:**
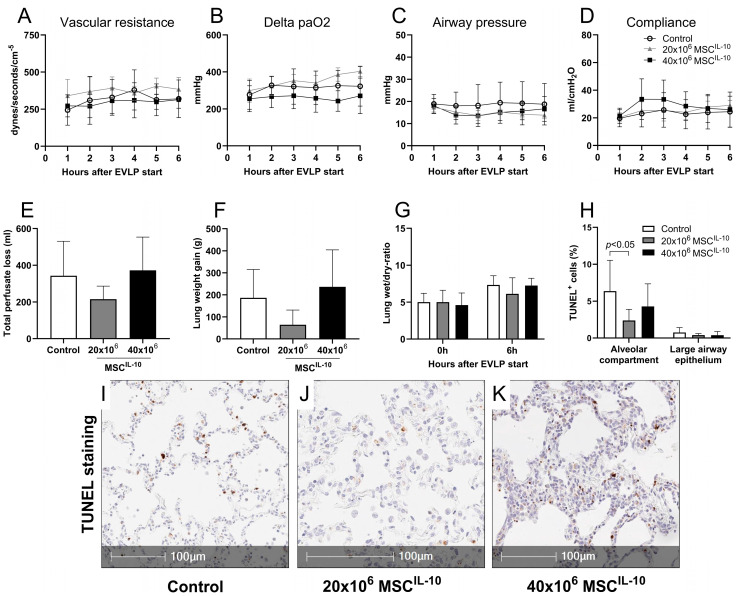
Lung function during EVLP after MSC^IL-10^ administration. Pig lungs with 24 h cold ischemia were connected to clinical-grade EVLP and randomized to control (*n* = 7), or to receive 20 × 10^6^ (*n* = 5) or 40 × 10^6^ MSCs^IL-10^ (*n* = 6) through the pulmonary artery 1 h after EVLP start. Lung function parameters were recorded every hour for 6 h. (**A**) Pulmonary vascular resistance, (**B**) delta pO2, (**C**) peak airway pressure and (**D**) dynamic compliance. Vascular permeability and lung edema were evaluated by (**E**) total perfusate loss and (**F**) lung weight gain during EVLP, and by (**G**) lung wet/dry ratio at the end of EVLP. Cellular apoptosis was evaluated from tissue sections at the end of EVLP by (**H**) TUNEL staining using automated image analysis and tissue segmentation to alveolar and large airway epithelium compartments. Representative TUNEL^+^ staining of (**I**) control, (**J**) 20 × 10^6^ and (**K**) 40 × 10^6^ MSC^IL-10^ lungs. Data mean ± SD, analyzed by 2-way ANOVA (**A**–**D**,**G**,**H**) or 1-way ANOVA with Dunnett’s correction (**E**,**F**) comparing treatment groups to the control group. EVLP, ex vivo lung perfusion; IL-10, interleukin-10; MSCs, mesenchymal stromal cells; PA, pulmonary artery; TUNEL, deoxynucleotide transferase-mediated deoxy uridine triphosphate nick-end labeling.

**Figure 4 cells-13-00859-f004:**
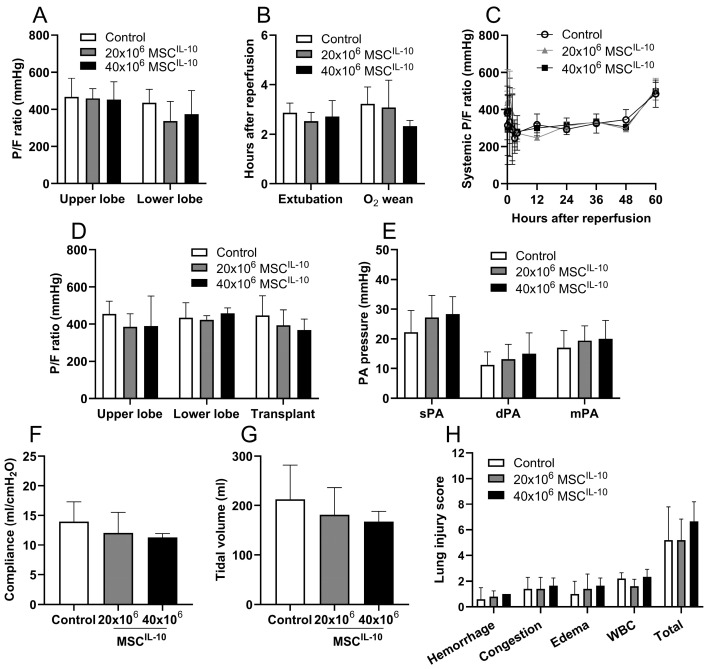
Effect of MSC^IL-10^ treatment on lung function up to 3 days after transplantation. Left single lungs, which had been randomly assigned to control (*n* = 5), or to receive 20 × 10^6^ (*n* = 5) or 40 × 10^6^ (*n* = 3) MSCs^IL-10^ during EVLP, were transplanted into recipient pigs with methylprednisolone and cyclosporine A background immunosuppression. (**A**) Early graft function was evaluated by measuring the P/F ratio from samples taken selectively from the upper or lower veins of the transplant 1 h after reperfusion. Postoperative recovery was assessed by the time from graft reperfusion to recipient (**B**) extubation or wean of supplemental oxygen, and by (**C**) determining the P/F ratio for up to 3 days after transplantation. Post-recovery P/F ratios were measured during spontaneous breathing of room air, and the 3-day values during anesthesia and mechanical ventilation. At 3 days after transplantation, (**D**) transplant upper and lower lobe P/F ratios were measured, and the transplant P/F ratio, (**E**) pulmonary artery pressures, (**F**) dynamic compliance and (**G**) tidal volume were assessed after elimination of the contralateral native lung blood flow and ventilation by clamping. (**H**) At 3 days, transplant acute lung injury score was determined by a transplant pathologist from histological sections assessing for air space hemorrhage, vascular congestion, edema/fibrin in the alveoli and the presence of infiltrating white blood cells. Each parameter was scored from 0 to 3, and a total acute lung injury score was calculated. Data mean ± SD, analyzed by two-way ANOVA (**A**,**B**,**D**,**E**) or by one-way ANOVA (**F**,**G**) with Dunnett’s correction, or in cases of missing values by mixed-effect analysis with Šidák correction (**C**), or non-continuous parameters by Kruskal–Wallis test with Dunn correction (**H**), comparing treatment groups to the control group. EVLP, ex vivo lung perfusion; MSCs, mesenchymal stromal cells; PA; pulmonary artery; sPA, systolic pulmonary artery pressure; mPA, mean pulmonary artery pressure; dPA, diastolic pulmonary artery pressure; P/F ratio, pO2 divided by the fraction of inspired oxygen; WBC, white blood cell.

**Figure 5 cells-13-00859-f005:**
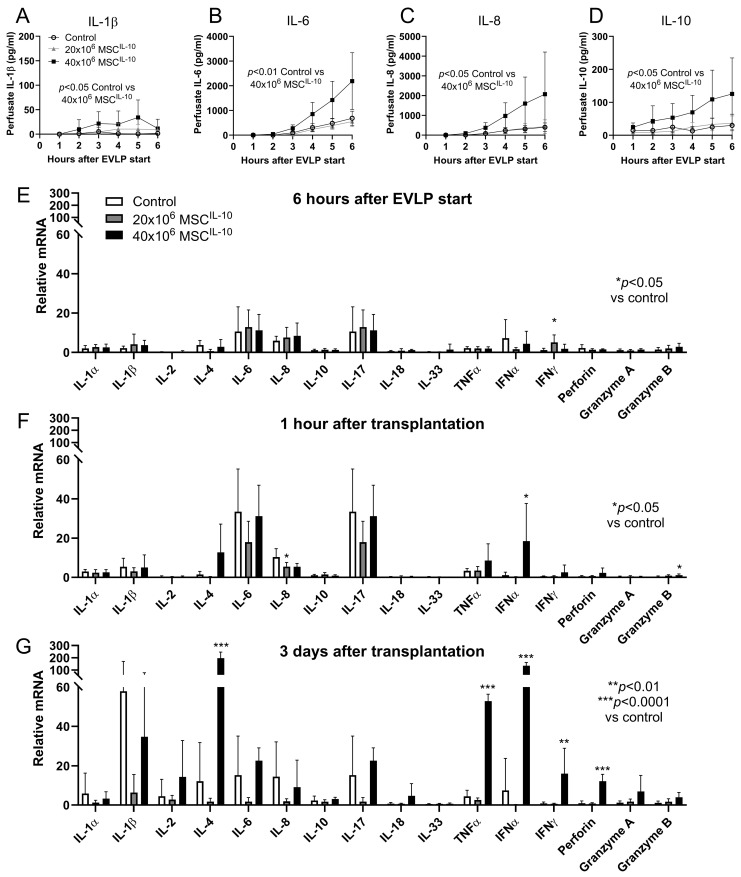
Cytokine profile after MSC^IL-10^ treatment. EVLP perfusate samples were collected every hour, and perfusate concentrations of (**A**) IL-1β, (**B**) IL-6, (**C**) IL-8 and (**D**) IL-10 were measured using pig cytokine-specific ELISA. Treatment with 40 × 10^6^ (*n* = 6), but not with 20 × 10^6^ MSCs^IL-10^ (*n* = 5), significantly increased perfusate cytokine concentrations compared to the control group (*n* = 7). Lung tissue cytokine mRNA profile was determined by qRT-PCR with pig-specific primers during (**E**) EVLP (*n* = 7, 5 and 6), and (**F**) 1 h (*n* = 6, 5 and 4) and (**G**) 3 days after transplantation (*n* = 5, 5 and 3 for control, 20 × 10^6^ and 40 × 10^6^ MSC^IL-10^ groups). qRT-PCR results were analyzed with the ddCt method using GAPDH as the housekeeping gene, and results are given in relation to respective pre-EVLP mRNA levels. Data mean ± standard deviation and analyzed by 2-way ANOVA (**A**–**D**), or by 1-way ANOVA (**E**–**G**), with Dunnett’s correction comparing treatment groups to the control group. EVLP, ex vivo lung perfusion; IFN, interferon; IL, interleukin; MSC, mesenchymal stromal cell; TNF, tumor necrosis factor.

**Figure 6 cells-13-00859-f006:**
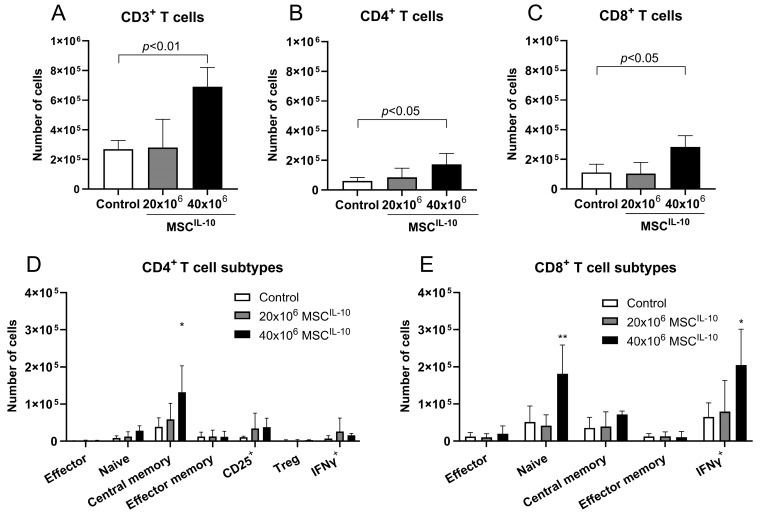
Lung transplant T cell composition after MSC^IL-10^ treatment 3 days after transplantation. T cells and their subtypes in lung transplants 3 days after transplantation were quantified by flow cytometry. Lung (**A**) CD3^+^, (**B**) CD4^+^ and (**C**) CD8^+^ T cells were increased in transplants that had received 40 × 10^6^ MSCs^IL-10^. T cell subtype analysis showed that treatment with 40 × 10^6^ MSCs^IL-10^ significantly increased (**D**) CD45RO^−^CD62L^+^ central memory CD4^+^ T cells and (**E**) CD45RO^+^CD62L^+^ naïve CD8^+^ T cells and IFNγ^+^ cytotoxic CD8^+^ T cells. Data mean ± standard deviation and analyzed by 1-way ANOVA with Dunnett’s correction comparing treatment groups to the control group. * *p* < 0.05, ** *p* < 0.01. IFN, interferon; MSC, mesenchymal stromal cell; Treg, regulatory T cell.

**Figure 7 cells-13-00859-f007:**
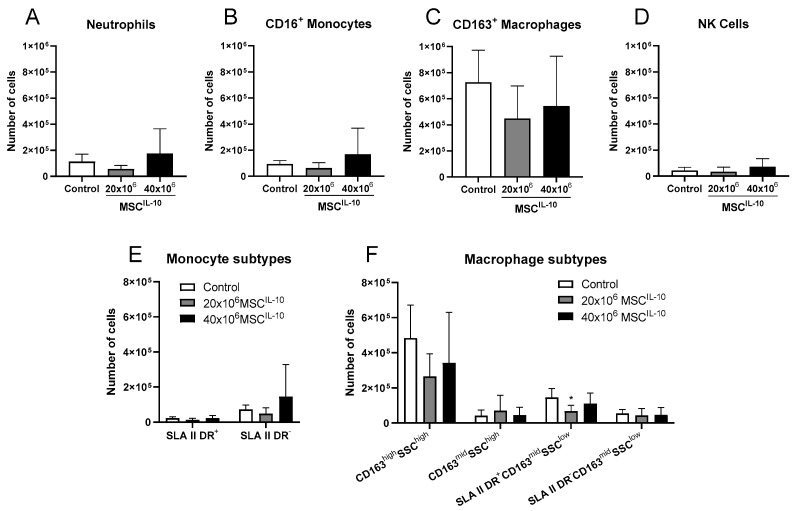
Lung transplant myeloid cell composition after MSC^IL-10^ treatment 3 days after transplantation. Myeloid cells and monocyte and macrophage subtypes in lung transplants 3 days after transplantation were quantified by flow cytometry. Lung (**A**) neutrophils, (**B**) CD16^+^ monocytes, (**C**) CD163^+^ macrophages and (**D**) NK cells. (**E**) Monocytes were further divided according to SLA II DR expression, and (**F**) macrophages according to CD163, side scatter and SLA II DR expression. Treatment with 20 × 10^6^ MSCs^IL-10^ significantly decreased the number of SLAII^-^DR^+^CD163^mid^SSC^low^-activated macrophages. Data mean ± standard deviation and analyzed by 1-way ANOVA with Dunnett’s correction comparing treatment groups to the control group. * *p* < 0.05. MSC, mesenchymal stromal cell; NK, natural killer; SLA, swine leukocyte antigen; SSC, side scatter.

**Figure 8 cells-13-00859-f008:**
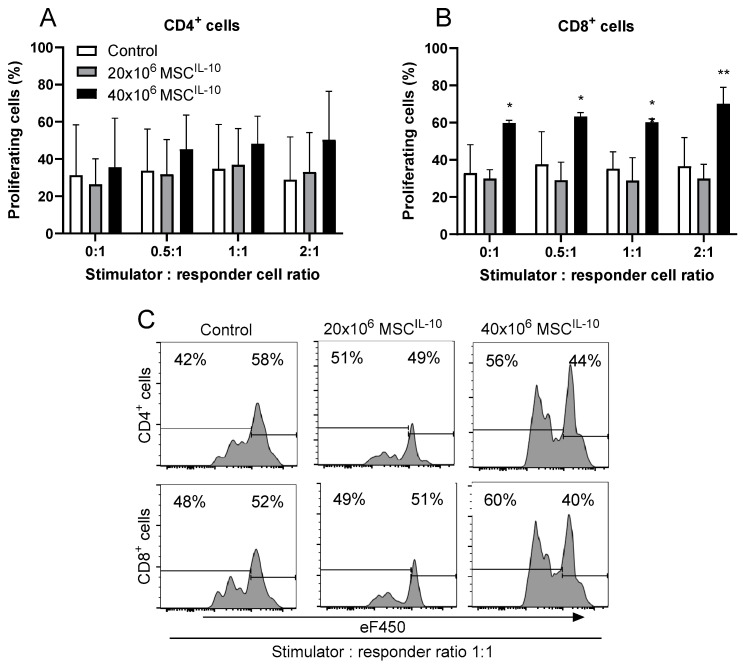
Effect of MSC^IL-10^ treatment on early alloimmune activation 3 days after transplantation. Early alloimmune response was determined using mixed lymphocyte reaction. Donor spleen cells were used as stimulators, and recipient spleen cells, procured 3 days after transplantation, were used as responders. CD4^+^ cell (**A**) and CD8^+^ cell (**B**) proliferation with different stimulators to responder cell ratios. Representative cell proliferation results of CD4 and CD8 cells with 1:1 stimulator: recipient cell ratio and eFluor 450 proliferation dye (**C**). Data mean ± standard deviation and analyzed by 1-way ANOVA with Dunnett’s correction comparing treatment groups to the control group. * *p* < 0.05, ** *p* < 0.01. MSC, mesenchymal stromal cell.

## Data Availability

The data that support the findings of this study are available from the corresponding author upon reasonable request.

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
