# Peer review of "Lung Transplant Immunomodulation with Genetically Engineered Mesenchymal Stromal Cells—Therapeutic Window for Interleukin-10"

_cells, 2024, doi:10.3390/cells13100859_

Round 1

Reviewer 1 Report

Comments and Suggestions for Authors

The authors employed a pig model of lung transplantation using EVLP to deliver genetically engineered mesenchymal stromal cells with augmented production of human anti-inflammatory IL-10 at different dosages. When compared with control animals there were no differences in graft function. Low-dose treatment induced a decrease of apoptosis and lower macrophage activity. High-dose treatment resulted in inflammation and cytotoxic CD8+ T-cell activation.

The authors are to be congratulated for an excellent example of hypothesis-driven research. The scientific question, methods and results are clearly presented. From my point of view the diverging results of the different dosages is rather short and could be discussed somewhat more in depth.

Author Response

The authors employed a pig model of lung transplantation using EVLP to deliver genetically engineered mesenchymal stromal cells with augmented production of human anti-inflammatory IL-10 at different dosages. When compared with control animals there were no differences in graft function. Low-dose treatment induced a decrease of apoptosis and lower macrophage activity. High-dose treatment resulted in inflammation and cytotoxic CD8+ T-cell activation.

The authors are to be congratulated for an excellent example of hypothesis-driven research. The scientific question, methods and results are clearly presented. From my point of view the diverging results of the different dosages is rather short and could be discussed somewhat more in depth.

We would like to thank the Reviewer for taking the time to review our work and for the constructive comments.

We have now expanded the Discussion to better highlight IL-10 levels achieved using the two the different MSCIL-10 doses in our study, and how the IL-10 levels correspond to previously published reports (Discussion, pages 14-15, lines 444-456).

Reviewer 2 Report

Comments and Suggestions for Authors

The manuscript “Lung transplant immunomodulation with genetically engineered mesenchymal stromal cells – therapeutic window for interleukin-10” by Nykänen et al. assess the use of engineered MSCs, overexpressing the anti-inflammatory cyokine IL-10, as a therapeutic intervation for preserving donor lungs. The topic of the manuscript if of utter importance, given the shortage of donor organs and the always increasing numbers of patients in the waiting list. The research is well conducted and the results are well presented. I have some comments for the authors.

Experimental design:

Why did the authors decided to use the therapeutic dose of 20 and 40 million cells per lung?

Why did the authors not include a control group with naive MSC? That’s the best control to evaluate differences between naive and engineered cells and to evaluate the impact of the IL-10 in the treatment.

Results and discussion:

Authors should add some images of the lungs architecture before and after the EVLP and after transplation, at least as a supplementary information. Moreover, I think that authors should analyze the presence of the different inmune cells in the tissue by inmunofluorescence.

The discussion could deepen more about the dosing of MSCs and also the amount of IL-10 that would have a desired therapeutic effect. Authors claim that after the high dose administration of MSCs, IL-10 levels were of about 17000pg/mL, which is rather high. What’s the physiological level of IL-10 or before the EVLP start? This may be linked to the undesired pro-inflammatory effects observed in this group.

Could the authors consider using naive MSCs in concomitance to recombinant IL-10 as a possible treatment?

Author Response

The manuscript “Lung transplant immunomodulation with genetically engineered mesenchymal stromal cells – therapeutic window for interleukin-10” by Nykänen et al. assess the use of engineered MSCs, overexpressing the anti-inflammatory cyokine IL-10, as a therapeutic intervation for preserving donor lungs. The topic of the manuscript if of utter importance, given the shortage of donor organs and the always increasing numbers of patients in the waiting list. The research is well conducted and the results are well presented. I have some comments for the authors.

Experimental design:

Why did the authors decided to use the therapeutic dose of 20 and 40 million cells per lung?

We decided to use 20 and 40 million cells per lung based on our previously published results of unmodified MSCs administered during pig EVLP, and on our pilot experiments using engineered MSCIL-10 cells.

Mordant et al (J Heart Lung Transplant 2016) investigated different doses and administration routes of umodified MSCs using a pig EVLP model. Intravenous route was better than airway delivery, and when comparing different intravenous MSC doses (50, 150 or 300 million cells),150 million cells was the highest tolerated dose, while 300 million cells resulted in incresed pulmonary artery pressure possibly due to intravascular MSC clustering.

As our pilot experiments with engineered MSCIL-10 cells indicated very efficient IL-10 production by the modified MSCs, we decided to use 20 and 40 million MSCIL-10 cells in our current experiments. We believed that using these cell doses would not increase pulmonary artery pressures of the treated lungs but would enable comparison of possible dose-related effects.

Why did the authors not include a control group with naive MSC? That’s the best control to evaluate differences between naive and engineered cells and to evaluate the impact of the IL-10 in the treatment.

This is an important point and we considered including a naive MSC group in our experiments. However, we opted to include only the present 3 groups (no cells vs 20 vs 40 million MSCIL-10 cells) as the effects of naive MSC cells during EVLP have been previously extensively investigated (Mordant et al J Heart Lung Transplant 2016 and Nakajima J Heart Lung Transplant 2019). Also, we were worried that including additional groups would decrease the statistical power of the study and we therefore decided to concentrate to document possible dose-related effects by using only MSCIL-10 cells.

Results and discussion:

Authors should add some images of the lungs architecture before and after the EVLP and after transplation, at least as a supplementary information. Moreover, I think that authors should analyze the presence of the different inmune cells in the tissue by inmunofluorescence.

We have now included representative images of lungs before, during and after EVLP, and 3 days after lung transplantation (Supplementary Figure S3).

As presented in Figures 6 and 7, we investigated the effects of MSCIL-10 cells on lung immune cells using FACS. Our FACS panels (Supplementary Figures S4 and S5) enabled us to not only compare the contribution of CD3+, CD4+, CD8+, neurtophils, monocytes, macrophages and NK cells in lung tissue, but also to investigate specific immune cell subtypes. Therefore, we believe that lung tissue immunofluorescence stainings of different immune cells would not provide additional information over the presently used lung tissue FACS.

The discussion could deepen more about the dosing of MSCs and also the amount of IL-10 that would have a desired therapeutic effect. Authors claim that after the high dose administration of MSCs, IL-10 levels were of about 17000pg/mL, which is rather high. What’s the physiological level of IL-10 or before the EVLP start? This may be linked to the undesired pro-inflammatory effects observed in this group.

We have now expanded the Discussion to better highlight the IL-10 levels achieved using the two the different MSCIL-10 doses in our study, and how the IL-10 levels correspond to previously published reports (Discussion, pages 14-15, lines 444-456).

EVLP perfusate pig IL-10 levels were 9 pg/ml at 1 hour (before administration of MSCIL-10 cells with augmented production of human IL-10) and 30 pg/ml at 6 hours after EVLP start. These low endogenous IL-10 levels are in line with our previous EVLP experiments and indicate that MSCIL-10 administration indeed resulted in profound elevation of human IL-10.

We agree that the very high IL-10 levels probably resulted in the undesired pro-inflammatory effects of the high-dose MSCIL-10 group. The expanded Discussion now provides some guidance for future studies that aim for anti-inflammatory and protective effects using lower IL-10 dose, or for cytotoxic effects with higher IL-10 levels.

Could the authors consider using naive MSCs in concomitance to recombinant IL-10 as a possible treatment?

We agree that combining naive MSCs and recombinant IL-10 would be interesting and future experiments that evaluate this approach would be warranted. As the combined cell and gene therapy used in the present study probably results in higher local tissue IL-10 levels than intravascular administration of recombinant IL-10, it remains to be seen whether these two strategies result in divergent outcomes.